# Possible Explanations for Rising Melanoma Rates Despite Increased Sunscreen Use over the Past Several Decades

**DOI:** 10.3390/cancers15245868

**Published:** 2023-12-16

**Authors:** Rebecca Lapides, Babak Saravi, Alina Mueller, Michael Wang-Evers, Lara Valeska Maul, István Németh, Alexander Navarini, Dieter Manstein, Elisabeth Roider

**Affiliations:** 1The Robert Larner, M.D., College of Medicine, University of Vermont, Burlington, VT 05405, USA; rebecca.lapides@med.uvm.edu; 2Cutaneous Biology Research Center, Department of Dermatology, Massachusetts General Hospital, Harvard Medical School, Charlestown, MA 02129, USA; mevers@mgh.harvard.edu (M.W.-E.); dmanstein@mgh.harvard.edu (D.M.); 3Department of Orthopedics and Trauma Surgery, Medical Center, Faculty of Medicine, University of Freiburg, 79106 Freiburg, Germany; babak.saravi@jupiter.uni-freiburg.de; 4Department of Dermatology, University Hospital Basel, 4055 Basel, Switzerland; alina.mueller@usb.ch (A.M.); laravaleska.maul@usb.ch (L.V.M.); alexander.navarini@usb.ch (A.N.); 5Department of Dermatology, University Hospital Zurich, 8091 Zurich, Switzerland; 6Faculty of Medicine, University of Zurich, 8006 Zurich, Switzerland; 7Department of Dermatology and Allergology, Szent-Györgyi Albert Medical School, University of Szeged, 6720 Szeged, Hungary; nemethistvanbalazs@gmail.com

**Keywords:** sun protection, sun exposure, sunscreen, cutaneous melanoma, reactive oxygen species

## Abstract

**Simple Summary:**

A clear understanding of why the incidence of cutaneous melanoma continues to rise despite the increased use of sunscreens within the last several decades is lacking. Given how aggressive cutaneous melanoma can be, the aim of this communication is to better elucidate the relationship between sunscreen use and melanoma development and if there are other preventative measures to be aware of. We summarize some of the limitations of existing studies that investigated sunscreen use and melanoma development and highlight the importance of performing new studies that minimize such limitations to obtain more reliable results. This communication is intended to emphasize the importance of continued research not only to determine the most important factors contributing to increasing melanoma rates but also to establish clear guidelines and recommendations to reduce the risk of melanoma development.

**Abstract:**

The incidence of cutaneous melanoma continues to rise despite the increased use of sunscreens within the last several decades. Some research even suggests that the use of sunscreen is associated with increased rates of melanoma. Given the aggressive, and often deadly, nature of cutaneous melanoma, the aim of this communication is to better elucidate the relationship between sunscreen use and melanoma development and if there are other preventative measures to be aware of. A search was performed to identify the studies that have investigated melanoma development in individuals who used sunscreen and those who did not. Study limitations and possible confounding variables were identified, which guided a subsequent search to determine what data were available to support that these limitations and confounding variables may explain the perplexing association between sunscreen use and melanoma development. Five hypotheses were generated, which were related to increased awareness and reporting, the relationship between sunscreen use and the duration of sun exposure, the importance of broad-spectrum protection, and the effect of sunscreen on reactive oxygen species formation. The main conclusion is that more recent studies that control for confounding variables are required to determine the true effect of adequate broad-spectrum sunscreen use today on the development of melanoma.

## 1. Introduction

Cutaneous melanoma accounts for 1% of all skin cancer cases; however, it can be dangerous and even deadly if not treated early. When melanoma is diagnosed in an early stage, a wide excision can be performed, and this is typically curative. However, with more advanced disease, especially metastatic spread, effective treatment becomes much more difficult. In fact, the 10-year overall survival for stage I melanoma has been reported as 98%, yet that of stage IV melanoma is 75% [1]. Thus, efficient diagnosis and treatment initiation are critical to optimize outcomes for patients with melanoma.

In the United States, from 2009 to 2019, the number of new melanoma cases is estimated to have increased by over 40%, and over the last three decades, the percentage of individuals who develop melanoma has more than doubled [2]. The climbing incidence rates of melanoma over approximately the past 20 years in the United States (USA), the United Kingdom (UK), Australia, Italy, and Germany are shown in Figure 1. The incidence continues to increase in the United States, as the projected number of new cases in 2030 is 110,000 compared to approximately 65,000 new cases in the year 2011 [3,4]. Improved screening and detection of melanoma in earlier stages may play a role in the increase in incidence [5]; however, other factors that likely play a role should be considered. In Germany, a skin cancer screening initiative was implemented in 2008 [6]. From January 2009 to December 2010, the average participation rate in the screening program reached 30.8% [7]. Interestingly, the introduction of this national screening program was closely followed by a 29% increase in the occurrence of cutaneous melanoma in both genders [8]. It is important to note that the data representation for the UK differs from that of the other countries shown. The UK data are presented in intervals (e.g., 2002–2004) while the other country data are reported for individual years. For Italy, the data were limited to 2012. To ensure comparable visualization across all datasets, we selected the last date of each interval (e.g., 2004) for the year tick on the graph, aligning the representation of the UK data with the individual-year reporting of the other countries. Additionally, we utilized age-standardized rates to ensure comparability across different populations. This choice of standardized rates allows for a more accurate comparison of trends between countries. Given that melanoma incidence is rising and that it can become a very aggressive and dangerous disease, determining ways to prevent melanoma development and progression, as well as informing individuals about these practices, is imperative.

Figure 2 shows the age-standardized mortality rates obtained from the International Agency for Research on Cancer (Cancer Over Time dataset). The age-standardized rates were adjusted to the world standard population. The ASR for melanoma in Germany showed a general increase in the earlier years, peaking around 2011, followed by a slight decrease toward 2018. The peak ASR was observed in the early 2010s, indicating an upward trend in melanoma mortality rates during this period, which subsequently began to decline. In Australia, there was a notable decrease in the ASR over the study period. The rates started higher compared to other countries but showed a significant downward trend, especially after 2008, reflecting effective melanoma management and prevention strategies. The ASR in Italy exhibited fluctuations throughout the period. There were periods of increase, notably, around 2014 and 2016, but the overall trend shows a slight decrease toward the end of the study period, especially after 2016. The UK’s ASR showed a gradual increasing trend until around 2014, followed by a slight decrease. The rates remained relatively stable compared to other countries, with modest fluctuations over the years. The USA displayed a consistent decrease in the ASR from 2004 to 2018. The rates were initially on the higher side but showed a steady decline, indicating improvements in melanoma treatment and prevention. Across all countries, there was a noticeable variation in the ASR, with some countries like Australia showing a significant decrease, while others like the UK and Italy exhibited more stable trends with slight fluctuations. Overall, although the incidence rates increase across the countries, the mortality rates seem to be stable or reveal a decreasing pattern. These findings might reflect the effectiveness of public health policies, healthcare systems, and awareness campaigns resulting in more diagnoses of early-stage melanoma as well as better treatment options and algorithms. 

Furthermore, our findings correspond with the observations made by Welch et al. [13], who noted that melanoma mortality has remained generally stable despite a steep rise in incidence rates. This stability in mortality, as Welch et al. suggest, could be indicative of a stable true occurrence of cancer, thus implying that the increasing incidence rates might be reflective of overdiagnosis. Notably, there has been a minor yet significant decline in mortality in recent years. This decline may not solely be attributable to early detection but also aligns with advancements in melanoma treatments, such as checkpoint-blockade immunotherapies and targeted therapies for metastatic melanoma. Additionally, the rise in the incidence of metastatic melanoma at first detection, as pointed out by Welch et al., might be influenced by more intensive diagnostic evaluations and the use of increasingly sensitive tests, such as positron-emission tomography combined with computed tomography, leading to the identification of metastases that were previously undetected. These insights provide a nuanced understanding of the trends observed in our study, highlighting the complexity of interpreting cancer incidence and mortality data in the context of evolving diagnostic and treatment landscapes.

The distribution of age-standardized melanoma incidence rates among the EU-27 for 2022, the latest data from the European Cancer Information System, is depicted in Figure 3. The highest rates are observed in northern countries, including Denmark (149.1), Sweden (136.0), the Netherlands (114.5), Finland (77.1), Ireland (72.9), and Belgium (67.8). Figure 4 presents the relative change (%) in age-standardized incidence rates compared to the EU-27 average, further substantiating the findings mentioned above.

The long-term projections of cancer incidence rate changes for the EU-27 countries, comparing 2040 with 2022, are illustrated in Figure 5. These estimates indicate an increase in melanoma cases within the EU-27, rising from 101,510 in 2022 to 115,540 in 2040. This represents a relative increase of 13.83%. Ireland is expected to see the most significant relative increase (45%), followed closely by Luxembourg (44.7%), Malta (43%), and Cyprus (31%). Latvia stands out as the only country where a decrease in melanoma rates is projected, at −1.7%.

The benefits of sunscreen use for skin protection are often advocated by primary care providers and dermatology professionals. There is strong evidence that supports that sunscreen use significantly reduces the risk of squamous cell carcinoma [15]. The evidence for sunscreen reducing the risk of basal cell carcinoma is not as strong; however, it has been suggested that sunscreen may reduce the rate of multiple occurrences of basal cell carcinoma [15]. The evidence is very unclear regarding sunscreen use and the development of melanoma, as there are even several studies that find a positive association between sunscreen use and melanoma development [16,17,18,19]. This is perplexing, given that the purpose of sunscreen is to reduce skin damage from ultraviolet exposure, which is an established contributor to the pathophysiology of melanoma development [20]. Further, it is well documented that the purchasing of sunscreen has increased since the 1990s, which suggests that an increasing number of individuals are using sunscreen at least occasionally, if not more often [21]. Figure 6 demonstrates the upward trend in sunscreen usage in the USA since 2004. Figure 7 depicts the increase in Google searches related to sun protection in the USA, the UK, and Germany. Notably, these sun-protection search terms peak in frequency during the summer months.

Notably, some have suggested that despite the increased purchasing of sunscreen products, non-melanoma skin cancer incidence has risen as well [21], which may also have to do with the increasingly aging population [24]. However, given the strong body of evidence for sunscreen use protecting against non-melanoma skin cancer, particularly squamous cell carcinoma, and supporting that untreated melanoma has a higher potential for mortality compared to non-melanoma skin cancer [25], the focus of this communication is on establishing a better understanding of the relationship between sunscreen use and melanoma development, as well as possible other factors that affect melanoma formation. This non-exhaustive article discusses five hypotheses that offer an explanation for why melanoma incidence continues to rise and why this trend has been demonstrated even in individuals who use sunscreen regularly and makes suggestions for directions for future research.

## 2. Materials and Methods

A search was performed using the PubMed and Google Scholar search engines to identify the most pertinent studies that have investigated melanoma development in individuals who use sunscreen and those who do not. Keywords and phrases used for the search included “sunscreen use and melanoma”, “sun protection and melanoma”, “melanoma risk factors”, and “prevention of melanoma.” Studies that focused on the following terms were excluded: artificial UV light, infrared rays, actinic rays, dysplastic nevus syndrome, choroid/uvea/mucosal melanoma, congenital/becker/dysplastic nevus, blue rubber bleb nevus, nevus flammeus, and minimal erythema dose. There were no limitations set on the date range for included studies, as many of the most relevant studies were performed many decades ago. Nine highly relevant studies were identified ranging in date from 1979 to 2002. 

Study limitations and possible confounding variables were identified after either being addressed directly in the study or through our own critical evaluation. A subsequent search was conducted to determine what data were available to support that these limitations and confounding variables may help to explain the perplexing association between sunscreen use and melanoma development. Keywords and phrases included “skin cancer prevention”, “sun protection and sun exposure”, “sunscreen use and sun exposure”, “broad-spectrum sunscreen labeling”, “sunscreen and ROS formation”, “ROS formation and melanoma”, “sunscreen and oxidative effects”, and “oxidative damage and melanoma.” Once this evidence was gathered, five hypotheses were generated that had the most supporting literature available.

## 3. Results

The five hypotheses are described here, which all offer an explanation for why melanoma rates continue to rise despite increased sunscreen use over the last several decades. 

### 3.1. Hypothesis 1

One potential explanation pertains to increased patient and physician awareness of cutaneous malignancies, including melanoma, which increases reports and documentation. This would suggest that increased documentation yields a more accurate reporting of the true incidence of melanoma. 

### 3.2. Hypothesis 2

It is well documented in the literature that sunscreen use increases the duration of sun exposure. Thus, it is plausible that with the increased sunscreen use over the recent decades, individuals have spent more time exposed to the sun. Further, increased sunscreen use does not mean that the sunscreen is being used in the necessary quantities or is being reapplied appropriately, which means that increased sun exposure in this context could be quite damaging. 

### 3.3. Hypothesis 3

Until 1990, many sunscreens did not adequately include sun filters capable of absorbing ultraviolet A (UVA) radiation, and it was not until 2011 that the Food and Drug Administration (FDA) began regulating the standard for “broad-spectrum” sunscreen labeling. This means that all studies conducted prior to 2011 that investigated sunscreen use and melanoma development were likely performed with sunscreens that did not provide the same level of protection that sunscreens on the market today do. This calls into question the reliability of many of these older studies that conclude that sunscreen use does not reduce the risk of melanoma.

### 3.4. Hypothesis 4

Sunscreens may contribute to reactive-oxygen-species-mediated DNA damage. In certain areas such as Europe, sunscreens are regulated only as cosmetics, which has called into question the safety of certain European UV filters. There are also studies that show that some chemical sun filters may induce ROS formation. It is possible that these effects have contributed to melanoma pathogenesis, which also emphasizes the importance of regulating sunscreen formulations and UV filters very closely. 

### 3.5. Hypothesis 5

Climate change has resulted in warmer weather conditions during times of the year that would otherwise have been colder. The changing climate increases melanoma risk directly via an increased UV index, and the warmer temperatures may encourage more time spent outside exposed to the sun during times of the year when individuals otherwise would have spent more time inside. 

## 4. Discussion

The first hypothesis that offers an explanation for the increasing incidence of melanoma over the recent decades is that there is increased patient and physician awareness of cutaneous malignancies, including melanoma, which, in addition to improved screening tools, increases reports and documentation. The field of dermatology is implementing the use of dermoscopy and confocal microscopy to detect skin lesions earlier [26,27,28]. These tools allow for improved screening compared to simply using the naked eye to identify suspicious lesions. Thus, more lesions are diagnosed but in an earlier stage. Further, when dermoscopic and confocal images are correlated with the histopathological images, the result is an improved representation of the lesion [29]. Thus, some histopathologists may now use these images to diagnose a melanoma, which would previously have been diagnosed as a moderately or severely dysplastic nevus in the past. However, earlier recognition and diagnosis increase incidence but improve the prognosis for such lesions. It is clear that over the past several decades, the incidence of melanoma has increased along with sunscreen usage. It is important to mention that it has been suggested that despite the increased incidence of melanoma, mortality has remained stable, which begs the question of whether overdiagnosis is a concern considering the improved screening modalities mentioned above [12]. There are a myriad of reasons for this, such as the lowered threshold of clinicians seeking further evaluation of a pigmented lesion, the fear of missing a melanoma, and patient anxiety. It is important to consider the possibility of overdiagnosis and take measures to implement guidelines that support clinicians in evaluating and diagnosing appropriately. 

Adding to the increase in awareness are the several prevention campaigns that were introduced in the early 1980s to combat skin cancer, which potentially were the source of increased awareness and, therefore, increased reports of melanoma [30,31]. Further, with the exception of Australia, these prevention campaigns were not accompanied by the introduction of sustainable sun protective measures, so awareness was increased, but tools to combat risk factors for skin cancer were not implemented. Australia, however, did introduce measures such as sun-protective clothing for children and the installation of sun sails in public areas. This highlights that sun protection extends beyond just sunscreen use and includes other behaviors such as sun avoidance and wearing protective clothing as well as hats and sunglasses while outside. With the introduction of these additional measures, the rise in skin cancer rates that was observed in other areas was not seen in Australia [24]. In fact, after 2005, the incidence rate of melanoma flattened and even began declining at a rate of about 0.7% per year, although an increase was present again after 2011 (see Figure 1) [32]. This suggests that increased awareness and documentation undoubtedly increase the incidence of a certain condition; however, increasing awareness while also implementing measures that mitigate risk factors for developing that condition helps control the rise in the number of cases. This is especially important given the evidence that increasing awareness of the risks of sun exposure may improve the use of sun protection, but it does not reduce melanoma-prone behavior, even among specialist healthcare professionals [33]. Thus, finding ways to increase awareness while mitigating risk factors and promoting preventative behaviors is the solution.

A focus should also be placed on the spread of awareness of the implications of sun exposure and melanoma development to individuals at much earlier ages, as it has been established that sunburns during childhood and adolescence significantly increase the risk of melanoma later in life and that this risk persists even with later increased sunscreen use [34]. Thus, a strong focus on increasing awareness in children, adolescents, and parents may help reduce risky behavior in the younger years so that rather than awareness in adulthood resulting in increased reporting in individuals that already have sun damage, awareness from a much earlier age will help mitigate risk factors and prevent melanoma development in future generations. 

The second hypothesis offers an explanation for why studies have found sunscreen use to be associated with the increased development of melanoma, which is that sunscreen use increases the duration of sun exposure. Many studies support that sunscreen use and the use of higher sun-protection factors increase the duration of sun exposure [35,36,37,38,39,40,41]. These studies did not measure sunscreen reapplication, which is recommended every two hours, so it is possible that individuals subjected themselves to unprotected sun exposure hours later after an initial application. Thus, the duration of sun exposure, especially unprotected sun exposure, may be a confounding variable in these studies that have found sunscreen use to be associated with higher melanoma rates. Further, many of these studies included sunscreens with a sun protection factor (SPF) lower than 30; however, 30 is the minimum SPF recommended by the American Academy of Dermatology for daily use [42], so this SPF or higher should be used in studies. Even when sunscreen with an adequate SPF is used, it is also well established that sunscreen users typically apply just a fifth to a third of the amount of sunscreen required to achieve the maximum SPF potential [35]. Researchers must be aware of this and develop a methodology for studies that controls for differences in the amount of sunscreen applied by participants. It is important to note that one study found that higher SPF did not influence sun exposure duration [43]. However, this study did not measure sunscreen reapplication, and it is plausible that individuals who prefer higher SPF for greater sun protection also prioritize regular sunscreen reapplication for maximum protection and other sun protective measures such as limiting sun exposure in total. Intermittent sun exposure due to the increase in vacation travels from north to south even in the winter may also contribute to the increased risk of melanoma [44]. It has been shown that reduced air-travel costs have significantly increased vacations by families with children during winter months, which has contributed to an increase in melanoma incidence due to the increased exposure of untanned skin to a large amount of sun during a time of the year when the skin would otherwise have likely been shielded from the sun [45]. Figure 8 underscores this trend by illustrating the increasing number of non-business-purpose trips by EU-27 residents to other foreign countries, both during the summer (June, July, August) and winter (December, January, February) months, over time. This upward trajectory in travel, particularly evident until the onset of the COVID-19 pandemic, aligns with the factors mentioned above that contribute to a heightened melanoma risk. Notably, after a significant dip in 2020 attributed to the pandemic, the data indicate a resurgence in travel activities. Summer trips have shown a recovery starting from 2020, and winter vacations have also begun to rebound since 2021. These patterns highlight the evolving nature of travel behaviors in the EU-27 countries and underscore the importance of continued awareness and research into the associated health implications, particularly regarding sun exposure and melanoma risk.

It is also important to note that sunscreens do not completely protect against all wavelengths of light [47], so even individuals who apply sunscreen regularly are not perfectly protected. Hence, direct exposure should still be minimized, not lengthened, as the above-referenced studies have demonstrated, for individuals who apply sunscreen. Thus, more studies are required with participants not only using sunscreen but also applying it in the necessary quantity to actually achieve the maximum sun protection promised by the product prior to sun exposure. 

The next hypothesis regarding the positive association between sunscreen use and melanoma is that many studies that associate sunscreen use with increased melanoma rates were performed when sunscreens did not have adequate broad-spectrum protection. Until 1990, most sunscreens did not include sun filters capable of absorbing ultraviolet A (UVA) radiation, which comprises 90–95% of sunlight, and we now know that UVA radiation plays a critical role in the development and progression of melanoma [48,49,50]. Moreover, it was not until 2011 that the FDA began regulating sunscreen labeling, effectiveness, and testing such that a standard was provided that needed to be met for “broad-spectrum”, meaning adequate UVA and UVB coverage, labeling [51]. Many of the studies found in the search that found sunscreen to be associated with melanoma development or that sunscreen did not protect at all against melanoma were performed well before 2011, ranging in year from 1979 to 2000 [17,52,53,54,55,56]. Thus, these studies are outdated, and more recent studies are required to determine the effect of adequate broad-spectrum sunscreen use today on the development of melanoma. Notably, there are some studies from the same time period that found sunscreen use to be protective against melanoma development [57,58,59,60]. Nevertheless, the controversy about this topic remains, and a consensus cannot be reached until new, high-quality studies are conducted. Further, studies must ensure that individuals using sunscreen use the appropriate amount and application techniques, as it has been shown that individuals often fail to apply the necessary amount of sunscreen to all sun-exposed areas, affording them lower protection than what is advertised on the product because they do not achieve the recommended 2 mg/cm^2^-thick layer [61,62]. With updated studies comparing melanoma development in individuals who use broad-spectrum sunscreen with correct application techniques to those who do not, a true consensus can be reached regarding the important relationship between sunscreen use and cutaneous melanoma. 

It is also worth noting that while both chemical and physical sunscreens are efficacious in preventing skin damage caused by UVA and UVB exposure [63], comparing each of these types of sunscreens to determine if one type is better at reducing melanoma risk is a direction for future research. Few studies have investigated consumer storage techniques for sunscreen products, as improper storage or using expired sunscreen products may diminish efficacy. Further, the use of cosmetic products that contain low SPF values may give individuals a false sense of adequate protection from the sun, leading to a reduced frequency and quantity of sunscreen use and reduced overall protection. Each of these factors plays a role in the adequacy of sun protection beyond broad-spectrum sunscreen use, and more research is needed to better elucidate individuals’ attitudes toward sunscreen storage and use with cosmetics. 

The next hypothesis is that certain sunscreens may contribute to reactive oxygen species (ROS) formation and oxidative damage to DNA, driving melanoma development. Certain sunscreens in certain areas may actually drive ROS formation, which can promote melanoma development secondary to oxidative damage to DNA [64,65,66]. In Europe, sunscreens are regulated as cosmetics, which are not regulated as rigorously as drugs are, so the safety of many of the EU-listed UV filters has been questioned. Also, several chemical sun filters were shown to induce ROS formation in aqueous solutions containing furfuryl alcohol [64,65]. These studies were published in the 1990s, which was the same time period in which the other studies discussed above were published that found sunscreen use to be associated with melanoma, so this may be another mechanism through which sunscreen formulations that were not regulated possibly contributed to melanoma pathogenesis. Ensuring that there are no ingredients in sunscreen products that may actually drive processes that can contribute to melanoma development is critical so that individuals can reap the benefits of the protection sunscreen offers with minimal risk. This further emphasizes the need for updated studies with the more tightly regulated sunscreens on the market today.

The final hypothesis is that the increase in melanoma incidence may be more related to factors other than sunscreen use, such as depletion in the ozone layer or variations in the UV index due to climate change [67,68]. Even with increased sunscreen use, these fluctuations can increase the risk of melanoma. Further, warmer temperatures may encourage individuals to spend more time outside, which would increase their overall sun exposure during a time of the year when they otherwise would have spent more time inside with minimal sun exposure. These factors must be considered as well, and the extent to which each of these factors influences the development of melanoma is a direction for future research. 

It is crucial to consider a balanced perspective. While our study highlights the apparent paradox of increased sunscreen use alongside rising melanoma rates, we acknowledge the complexities involved in this relationship. Welch et al. have noted the steep increase in melanoma incidence, which they attribute to factors beyond just sun exposure, including heightened awareness, improved diagnostic capabilities, and, possibly, overdiagnosis [12]. This perspective aligns with our findings and suggests that the rise in melanoma rates may not be due solely to the ineffectiveness of sunscreens. Instead, it points toward a multifaceted issue where increased detection, particularly of less aggressive melanomas, plays a significant role. Furthermore, the efficacy of sunscreens in preventing melanoma, particularly in its early stages, remains a topic of ongoing research [69]. While sunscreens are effective in blocking harmful UV radiation, their role in preventing melanoma is influenced by various factors such as the application frequency, the amount used, and the spectrum of UV radiation they cover. In light of these considerations, we aimed to present a comprehensive view of the data, acknowledging both the potential benefits of sunscreen in melanoma prevention and the complexities surrounding its relationship with melanoma incidence rates. Excluding the possibility of sunscreen ineffectiveness or the insensitivity of melanocytic lesions to sunscreens would be an oversight. Hence, we emphasize the need for further research in this area to fully understand the dynamics of melanoma incidence, sunscreen use, and other contributing factors. The provided data contribute to the ongoing dialogue in this field, highlighting the need for a nuanced understanding of melanoma epidemiology, where sunscreen use is one of many factors to be considered. This approach underscores the importance of comprehensive preventative strategies, including but not limited to sunscreen use, in the fight against melanoma.

## 5. Conclusions

Though melanoma is uncommon compared to other types of skin cancer, it is aggressive and often deadly if not treated in an efficient manner. Thus, its rising incidence over the last several decades is alarming, and more research is necessary to determine the best ways to prevent its development. The five hypotheses described here offer an explanation for not only why melanoma incidence has been rising but also why sunscreen use has not been shown to be protective against melanoma by several past studies. This has been perplexing, as sunscreen use has been shown to be highly effective at preventing other types of skin cancer. Regardless of which hypothesis, or perhaps a combination, is most accurate, more studies must be conducted that not only use regulated broad-spectrum sunscreen but also control for varying application techniques that may compromise the ability of the sunscreen to protect the skin to the fullest extent. Only once we have this updated data will we be able to reach a more reliable consensus on the impact of sunscreen on melanoma development. This relationship is crucial to elucidate so that we can accurately educate individuals about the optimal skin-protective measures to prevent this dangerous type of skin cancer and improve overall skin health and protection.

## Figures and Tables

**Figure 1 cancers-15-05868-f001:**
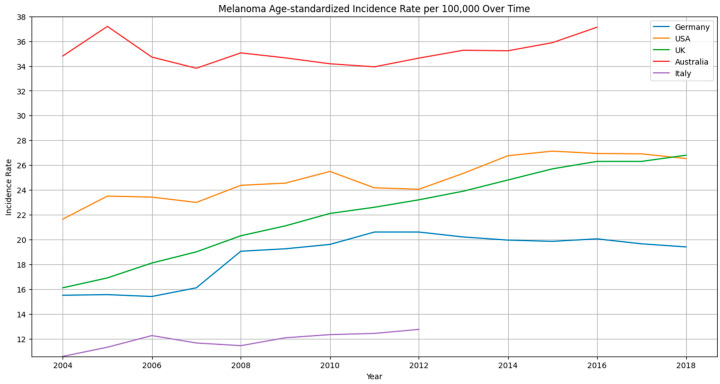
Age-standardized incidence rates (ASRs) per 100,000 for Germany [9], USA [10], UK [11], Australia [12], Italy [12]. Note: No data on ASR available for Italy from International Agency for Research on Cancer (Cancer Over Time dataset) after 2012. For the UK, data points in the source represent intervals (e.g., 2002–2004) rather than individual years as in the USA and Germany. The last date of this interval for each year’s x-axis tick was chosen for comparable visualization.

**Figure 2 cancers-15-05868-f002:**
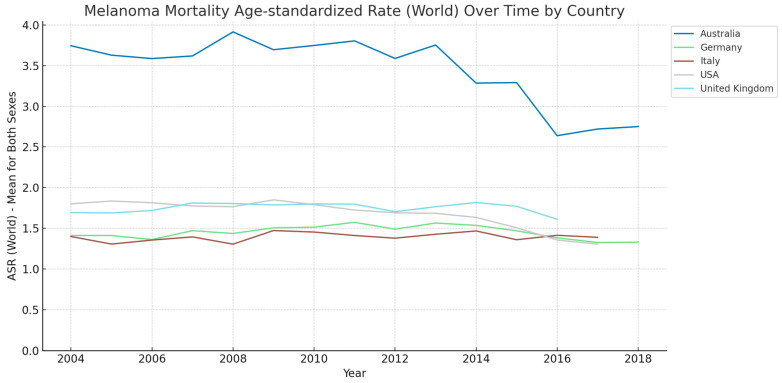
Age-standardized mortality rate per 100,000 (world normal population). Melanoma of skin: Australia—Germany—Italy—United Kingdom—USA [11]. Lines were smoothed using the LOESS regression algorithm (bandwidth: 0.25).

**Figure 3 cancers-15-05868-f003:**
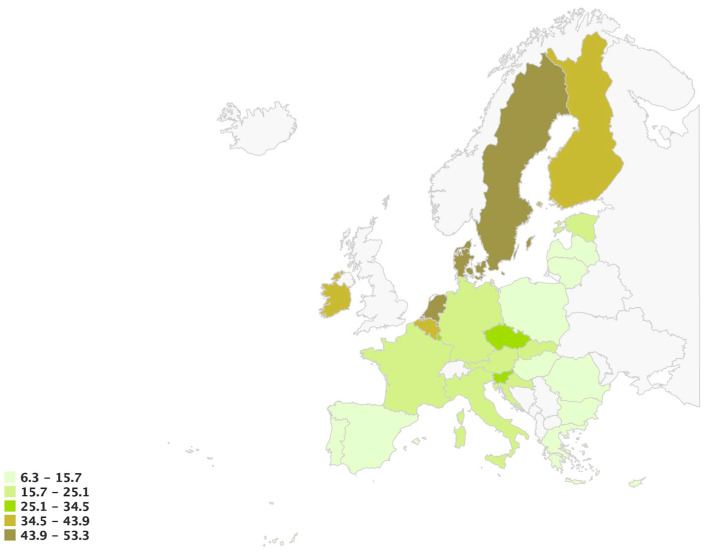
Age-standardized incidence rate per 100,000 in 2022 in EU-27. Melanoma of skin; both sexes, all ages [14].

**Figure 4 cancers-15-05868-f004:**
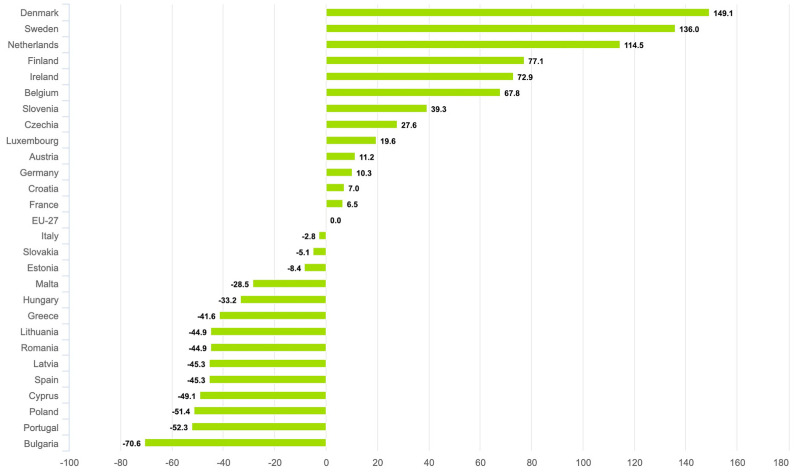
Age-standardized incidence rate per 100,000 in 2022. Relative changes (in %) of each country compared to EU-27. Melanoma of skin; both sexes, all ages [14].

**Figure 5 cancers-15-05868-f005:**
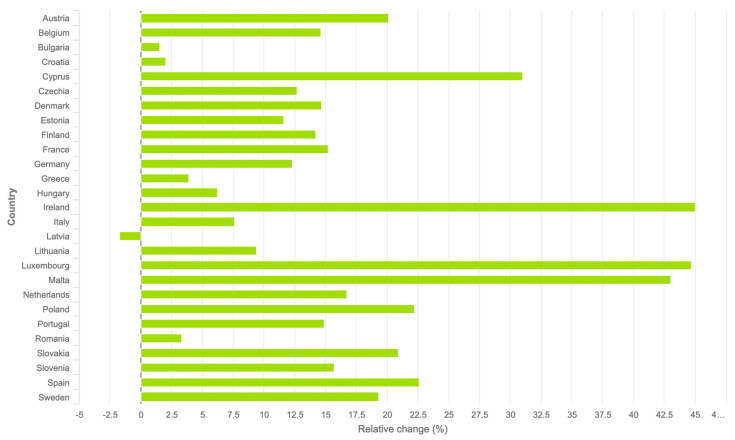
Age-standardized incidence rate per 100,000 projection 2040 compared to 2022. Relative changes (in %) of each country. Melanoma of skin; both sexes, all ages [14].

**Figure 6 cancers-15-05868-f006:**
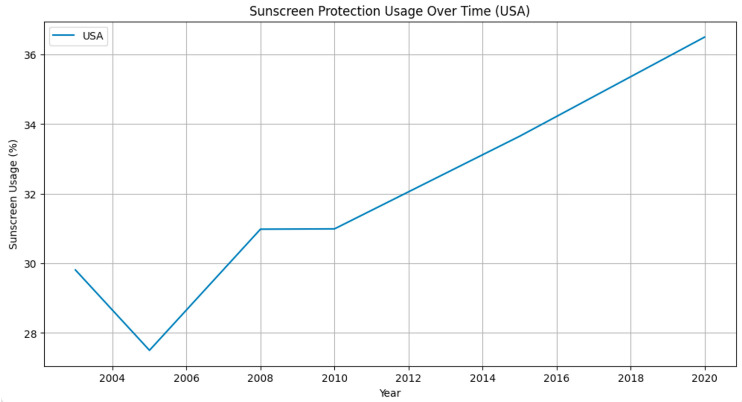
Rise in sunscreen usage in the USA from 2004 to 2020. Note: Percentage of adults aged 18 years and older (USA) who always or most of the time protect themselves from the sun by using sunscreen with a sun protection factor of 15 or higher [22].

**Figure 7 cancers-15-05868-f007:**
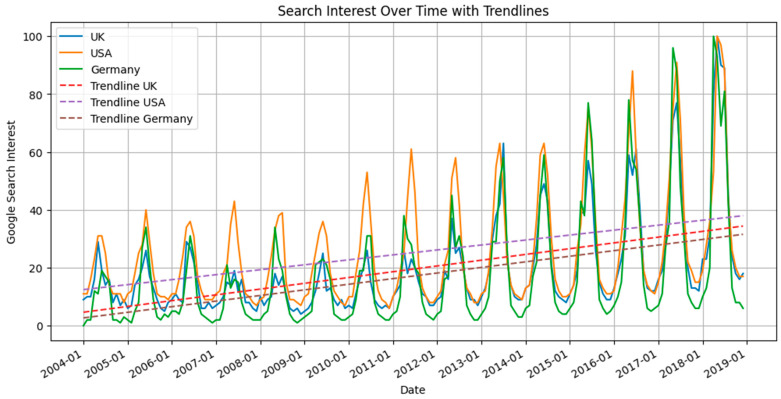
Comparative analysis of Google search interest related to sun-protection usage from January 2004 to December 2018 in the USA, the UK, and Germany. Data were sourced from Google Trends using the search terms ‘sunscreen’ for the USA and UK and ‘sonnencreme’ for Germany. Solid lines indicate actual search interest over time, while dashed lines depict trendlines for each respective region. The values represent search interest relative to the highest point on the chart for the given region and time period. A value of 100 denotes peak popularity for the term; 50 indicates that the term is half as popular, and 0 means insufficient data for the term. It is evident that search terms were used more frequently during summer months and have been increasingly searched over time, indicating a growing interest in sun-protection usage [23].

**Figure 8 cancers-15-05868-f008:**
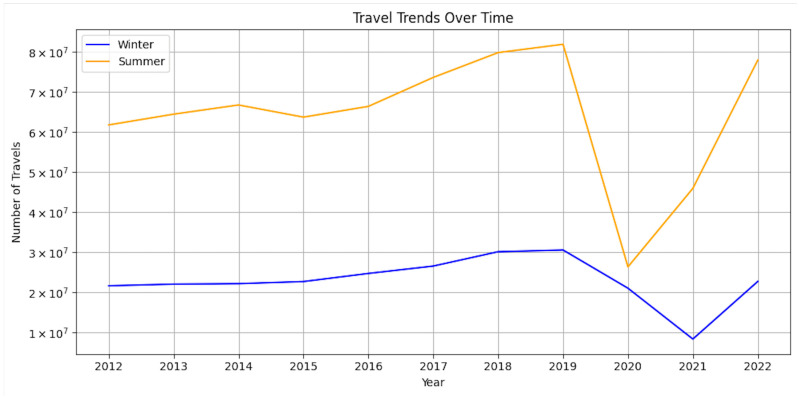
Non-business-purpose trips of EU residents (EU-27) to other foreign countries in summer (June, July, August) and winter (December, January, February) months. Minimum duration of vacation 4 nights and over [46].

## Data Availability

No new data were created.

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
