# Peer review of "Possible Explanations for Rising Melanoma Rates Despite Increased Sunscreen Use over the Past Several Decades"

_cancers, 2023, doi:10.3390/cancers15245868_

Round 1

Reviewer 1 Report

Comments and Suggestions for Authors

Overall a good paper on an interesting topic. Some improvements are recommended:

1 Line 30-31. Add reference to stated recent increased melanoma rates.

2 Line 41-42. The sentence has the word "that" twice and needs a minor tidy up to improve ease of reading.

3 Line 47. Needs correction: Merkel Cell Carcinoma is more dangerous than melanoma. Melanoma is not the most dangerous skin cancer.

4 Figure(s) 1. The word "incidence" implies melanoma presence. Melanoma reported is more accurate as not all melanomas can be diagnosed at initial presentation or even later presentations (the cases may be too small, too feature poor or beyond the resolution limits of the clinician).

5 There is no mention of the use of dermoscopy or confocal microscopy in early melanoma detection. The use of these instruments has been proven to increase pick up rates for early melanoma in many recent studies. Screening is mentioned in the text. However, there is a substantial difference in melanoma diagnosis rates when comparing naked eye screening with screening using dermoscopy.  

6 Line 308. Melanoma is not "more rare" compared to non melanoma skin cancers. Suggest "more rare" replaced by "uncommon".

7 Sun protection has a broad scope compared to sun screen use alone. Total sun protection involves/includes: sun avoidance behaviours, protective clothing cover, wearing hats and sun glasses etc when outside. These things can be confounders not mentioned in studies.

8 Finally, the histopathological diagnosis of melanoma is improving over the past 2 decades - when clinicopathological correlation is fully exploited -  including correlating dermoscopy and confocal images with the histology slides. What one histopathologist may cell a moderate or severe dysplastic naevus case may be called a melanoma by a different histopathologist examining the same case slides. We as a profession are tending to diagnose a higher proportion of smaller early melanoma now compared to many years ago. These cases obviously have a better prognosis.     

Comments on the Quality of English Language

Quality of English quite good.

Reviewer 2 Report

Comments and Suggestions for Authors

The manuscript entitled “Possible Explanations for Rising Melanoma Rates Despite In- 2 creased Sunscreen Use Over the Past Several Decades” presents data on melanoma incidence in three countries and its correlation with sunscreen use and offers four hypotheses to explain the apparent paradox of increased sunscreen use being associated with increased melanoma incidence. This is an important public health issue since preventative measures should be based on clear epidemiological evidence, which, in the case of melanoma and sunscreen is, at present, not the case.

Major remarks

The study has a central flaw that consists in the omission of data on mortality. The central question of whether increasing melanoma incidence is associated with increasing mortality is not addressed. To this end, the article by Welch et al. (NEJM 384,1,72,2021) should be considered.

Rather than reporting the results of the study and critically discussing them, the authors appear to be biased towards the explanation that sunscreens are effective and the apparent association with melanoma incidence must be explained excluding the possibility of ineffective sunscreens or insensibility of initial melanocytic lesions to sunscreens.  

Incidence rates in USA and Germany are discontinuous. This is unlikely to be due to a really oscillating incidence rather than methodological aspects that should be discussed.

Hypothesis 3: The normative changes in Europe must be cited since the article also shows data on UK and Germany.

The possibility that increased awareness and eventually economic interests might lead to overdiagnosis of melanoma or misdiagnosis of benign non-melanoma lesions as melanoma (Welch et al. as cited). Just like in all early diagnosis screenings, many early stage malignant lesions are identified that would, however, not have grown out to a clinically relevant lesion. Mis- and overdiagnosis would better be cited as an additional hypothesis.

Changed exposure habits due to winter travel should be considered. This is mentioned by just one sentence in the discussion section. In the last decades, there is a growing trend towards winter travel to destination with heavy sun. This also includes families with children. This out of season exposure of completely untanned skin to heavy sunshine might act as an additional and growing risk factor. This can easily be controlled by including travel statistics.

Climate change, cited in the last paragraph of the discussion, might also contribute to increasing melanoma incidence either directly or indirectly, since more warm days likely increase sun exposure in normally not so warm countries. This is an independent hypothesis.

The data for Australia should be shown in a fashion similar to those from the other countries.

For the sake of comparison, data from countries with strong sun exposure and good tumor registries, such as from South-Europe, should be included.
